# Vaccination Strategies Based on Bacterial Self-Assembling Proteins as Antigen Delivery Nanoscaffolds

**DOI:** 10.3390/vaccines10111920

**Published:** 2022-11-13

**Authors:** Félix Lamontagne, Vinay Khatri, Philippe St-Louis, Steve Bourgault, Denis Archambault

**Affiliations:** 1Department of Biological Sciences, Université du Québec à Montréal, C.P.8888, Succursale Centre-Ville, Montréal, QC H3C 3P8, Canada; 2Department of Chemistry, Université du Québec à Montréal, C.P.8888, Succursale Centre-Ville, Montréal, QC H3C 3P8, Canada; 3The Swine and Poultry Infectious Diseases Research Centre (CRIPA), Saint-Hyacinthe, QC J2S 2M2, Canada; 4Quebec Network for Research on Protein Function, Engineering and Applications (PROTEO), Quebec, QC G1V 0A6, Canada; 5The Center of Excellence in Research on Orphan Diseases—Fondation Courtois (CERMO-FC), Montréal, QC H3C 3P8, Canada

**Keywords:** vaccines, antigen delivery systems, nanostructures, bacterial self-assembling proteins, immunomodulation, self-assembly

## Abstract

Vaccination has saved billions of human lives and has considerably reduced the economic burden associated with pandemic and endemic infectious diseases. Notwithstanding major advancements in recent decades, multitude diseases remain with no available effective vaccine. While subunit-based vaccines have shown great potential to address the safety concerns of live-attenuated vaccines, their limited immunogenicity remains a major drawback that still needs to be addressed for their use fighting infectious illnesses, autoimmune disorders, and/or cancer. Among the adjuvants and delivery systems for antigens, bacterial proteinaceous supramolecular structures have recently received considerable attention. The use of bacterial proteins with self-assembling properties to deliver antigens offers several advantages, including biocompatibility, stability, molecular specificity, symmetrical organization, and multivalency. Bacterial protein nanoassemblies closely simulate most invading pathogens, acting as an alarm signal for the immune system to mount an effective adaptive immune response. Their nanoscale architecture can be precisely controlled at the atomic level to produce a variety of nanostructures, allowing for infinite possibilities of organized antigen display. For the bottom-up design of the proteinaceous antigen delivery scaffolds, it is essential to understand how the structural and physicochemical properties of the nanoassemblies modulate the strength and polarization of the immune responses. The present review first describes the relationships between structure and the generated immune responses, before discussing potential and current clinical applications.

## 1. Introduction

Vaccination has saved billions of lives and has significantly reduced the economic burden of many infectious diseases [1]. In addition to protecting the immunized individual from severe symptoms, vaccination confers protection to the community by limiting the spread of the targeted pathogen [1]. Despite the numerous advances in vaccine technologies over the last two centuries (Figure 1), a multitude of infectious diseases remain without any available vaccine. The human immunodeficiency virus (HIV), the respiratory syncytial virus (RSV), and many others have no clinically approved vaccine [2,3]. Moreover, several currently available vaccine formulations need to be optimized, including the Bacille Calmette-Guérin (BCG) vaccine and the seasonal flu vaccine. The BCG is still used to prevent infection with *Mycobacterium tuberculosis*, which causes tuberculosis. This formulation is effective in children for the first dose; however, the induced immunity diminishes over time, and it is not effective in booster doses [4]. Furthermore, there is a constant need to adjust the formulation of the influenza vaccine each year to accommodate antigenic drift and/or the major circulating strains, which can be challenging to predict [5]. This was exemplified by the 2014–2015 formulation, which showed an effectiveness of only 19% against infection [6].

Historically, vaccines were composed of whole pathogens that were either attenuated or inactivated. Attenuated vaccines are usually produced by a series of cell passages under suboptimal conditions to select variants with lower virulence in humans, whereas inactivated vaccines are produced by physical and/or chemical treatments [7]. Although live-attenuated vaccines induce a robust immune response, the pathogen can still replicate and mutate within the host, which risks its reverting to the original virulent form [8]. In contrast, inactivated vaccines are safer but tend to induce a moderate immune response, particularly in children and the elderly [9]. To address these limitations, vaccine technologies have shifted towards nucleic acid and protein-based vaccines (Figure 2) that aim to induce a targeted and safe immune response [10,11,12,13]. For instance, large-scale vaccination against SARS-CoV-2 has shown the efficacy and safety of nucleic acid-based vaccines. Cominarty (Pfizer-BioNTech) and Spikevax (Moderna, Cambridge, MA, USA), as well as Covishield/Vaxzevria (Oxford-AstraZeneca), have been administered to more than 69% of the global population (received at least one dose) [14]. This type of vaccine formulation delivers the antigenic coding sequence(s) to the host organism that expresses the protein(s) against which an immune response is to be induced [15]. To date, several nucleic acid-based vaccine technologies have been developed, such as mRNA, self-replicating RNA and plasmid DNA vaccines as well as various viral vectors [15]. Although they proved to be effective, these vaccine formulations usually require cold environments for long-term storage (from −20 to −80 °C), limiting its worldwide distribution [16]. In contrast, the protein-based subunit vaccines that have been used in clinics for many decades are stable formulations, although they tend to be weakly immunogenic on their own, requiring the use of adjuvants and/or immunomodulating delivery systems. The present review focuses on self-assembling bacterial proteins as nanoscaffold for antigen delivery in subunit vaccines.

## 2. Protein-Based Subunit Vaccines

Subunit protein vaccines are composed of one or more protein antigens to which a specific immune response is desired. The first subunit-based vaccine, Recombivax-HB (Merck, Rahway, NJ, USA), was approved in 1986 for use in humans [17]. It is composed of the hepatitis B virus (HBV) surface antigen, HBsAg, produced in the yeast Saccharomyces cerevisiae, which is mixed with aluminum salts (Alum), a commonly used adjuvant [17]. This vaccine formulation has demonstrated a protective efficacy of nearly 95% [18]. Since then, several protein-based subunit vaccines against viruses have been approved worldwide. Two other HBV vaccines, Heplisav-B (Dynavax, Emeryville, CA, USA) and Engerix-B (GlaxoSmithKline (GSK), Brentford, UK) have subsequently been licensed, with a similar efficacy to Recombivax-HB [18,19]. Other examples of protein subunit vaccine formulations approved for human use include Gardasil (Merck) and Cervarix (GSK) human papillomavirus (HPV) vaccine, Shingrix (GSK) varicella-zoster virus (VZV) vaccine, and Flublok (Protein Sciences Corporation) influenza A virus (IAV) vaccine [20]. Protein subunit vaccines are safe since they do not contain the original pathogen and they do not require highly specialized infrastructures for production or specialized equipment for long-term storage [21]. In order to produce the antigen, the coding gene is cloned into a vector, which is then transferred into a host cell for expression [22]. To date, five main expression systems have been utilized to produce recombinant proteins for vaccine purposes: bacterial, yeast, insect and mammalian cells, and more recently plants [22,23], with each of them having their advantages and limitations, as shown in Table 1.

## 3. Cellular and Molecular Mechanisms of Immune Responses to Subunit-Based Vaccines

The immune response to subunit vaccines is multifaceted. After the administration of the antigen(s) into the host organism, antigen-presenting cells (APCs), including dendritic cells, macrophages and monocytes, internalize the proteins and display them at their surface with the aim of activating the adaptive immune system (T and B lymphocytes) to mount an antigen-specific adaptive immune response. The activation of naive T and B cells occurs in secondary lymphoid organs (SLOs) such as the spleen or lymph nodes. To mount an effective immune response following vaccination, antigens must accumulate at these sites via cell-mediated or lymphatic transport [28].

Following antigen uptake, APCs are activated via a multitude of immune receptors (i.e., toll-like receptors (TLR) or cytokine receptors) [29]. They progressively gain the ability to present a high density of antigen-derived peptides and also upregulate co-stimulatory molecules that are necessary for the efficient activation of the adaptive immune response [30]. APCs migrate from peripheral tissues to SLOs and enter the T-cell zone (Figure 3), seeking to engage with a T cell receptor (TCR) that can recognize and bind to antigen-derived peptide loaded on type I or type II major histocompatibility complex (MHC) molecules presented at its surface.

Once a T cell is activated, it proliferates and produces a multitude of clones to help defend the organism against potential invaders. CD4 T cells or helper T cells (Th cells) bind to MHC-II-loaded peptides and secrete cytokines and express co-stimulatory molecules that help to activate other immune cells while polarizing the immune response [31]. Th cells orchestrate and direct the immune response toward a specific type of pathogen. Accordingly, there are multiple Th subsets: Th1 cells are associated with intracellular pathogens (i.e., viruses or certain bacteria), Th2 cells are mainly associated with helminth or allergen response, while Th17 cells induce protection against extracellular bacteria or fungi. T follicular helper cells (Tfh) are a specialized subset of CD4 T cells that localize to the B cell follicles in SLOs and promote B cell activation, germinal center (GC) reaction, antibody affinity maturation and isotype switching. The nature and the quality of the activating signals during antigen presentation greatly affect the polarization of Th cells and the resulting immune response [32,33,34]. Another type of T cells, CD8 T cells or cytotoxic T cells (CTL), recognize MHC-I-loaded peptides and are involved in killing infected cells. Although subunit vaccines are not efficient at promoting primary CTL response, they can effectively stimulate secondary CD8 T cells responses to common pathogens [35].

Simultaneously, B cells recognize, via their B cell receptor (BCR), membrane-bound antigens on follicular dendritic cells (FDCs) and subcapsular macrophages or soluble antigens that are drained to the follicles of the SLOs. After binding to antigens, the BCR and its bound antigen are internalized and digested before the antigen-derived peptide is docked on an MHC-II molecule at the surface of the B cell. This permits the interaction with an antigen-specific CD4 T cell that provides the co-stimulatory signal for full activation of the B lymphocyte. B cells can also be directly activated by antigens in a T-cell-independent manner following the strong signaling of co-receptors (i.e., TLR) or cross-linking of BCR by multivalent antigens [36].

Activated B cells start to proliferate and can take one of three differentiation paths. They can become short-lived plasma cells (SLPCs), or plasmablasts, a subset of B cells that secrete relatively low-affinity antigen-specific antibodies of switched or unswitched isotypes that defend the organism against immediate danger. Antigen-experienced B cells can also become GC B cells, which could enter the GCs, where they proliferate and undergo somatic hypermutation (SMH) to give rise to high-affinity antigen-specific BCR. Ultimately, GC B cells differentiate into long-lived plasma cells (LLPCs), and memory B cells (MBCs), or re-enter the GC for another round of proliferation and SMH. It is important to note that MBCs and LLPCs can arise through GC-independent mechanisms but with relatively low-affinity BCR [37,38]. After the initial proliferation burst following antigen recognition and GC reaction, LLPCs mostly migrate to the bone marrow, where they will secrete antibodies of high affinity following extensive SHM. On the other hand, MBCs take up residency in SLOs and other tissues in which antigen encounter is promoted. In these strategic locations, MBCs are in a quiescent state, ready to react to eventual re-exposure to the antigens [37].

## 4. Strategies to Enhance the Immune Response to Subunit Vaccines

Although protein-based subunit vaccines have shown efficacy in clinics, they remain poorly immunogenic when composed solely of soluble antigens. In fact, low-molecular-weight polypeptidic antigens are readily eliminated from the organism and generate little to no immune response. To circumvent these issues, different strategies have been developed, including (i) the use of adjuvants in the vaccine formulation, (ii) the addition of TLR agonists, and (iii) the conjugation of the antigen(s) to nanoparticles.

### 4.1. Adjuvants and Recruitment of Immune Cells at the Injection Site

Adjuvants are substances that can enhance, or modulate, the immune response directed toward an antigen. Traditionally, adjuvants were developed empirically, without a clear understanding of the molecular mechanisms involved in their immunostimulating properties [39]. Insoluble aluminum salts, or alum, remains the most widely used adjuvant in human vaccine formulations. Although its mechanisms of action are not fully understood, it is known that alum mainly works through the adsorption of antigens on particles of aluminum and the induction of an inflammatory milieu following the release of danger signals by affected cells. The pro-inflammatory environment caused by alum enhances the recruitment of immune cells, mostly neutrophils, at the site of injection and the antigen adsorption to particles increases uptake by APCs. This leads to an increase in antigen-directed antibodies. Alum also induces a CD4 T cell response that is Th2-biased in mice, while this bias is less clear in humans [40,41,42].

Oil-in-water emulsions are another type of adjuvant that is approved in humans and has been extensively used in vaccination. MF59 and AS03 are two proprietary adjuvants of Novartis and GlaxoSmithKline, respectively. Following the administration of antigen with an oil-in-water adjuvant, immune cells such as macrophages, dendritic cells and granulocytes are recruited at the site of injection. This leads to an increase in antigen uptake by APCs and greatly enhances the antibody and cellular immune response toward the antigen. Oil-in-water emulsions usually lead to a broader immune response and a more balanced Th1-Th2 cellular immune response compared to alum [39,41,43].

### 4.2. Stimulation of Immune Cells via the Activation of TLRs

Another strategy developed to increase the immune response emerged following the discovery of pattern recognition receptors (PRRs), which are germline-encoded immune receptors that bind to the pathogen-associated molecular pattern (PAMP) and danger-associated molecular patterns (DAMP). The PRRs are expressed on innate immune cells, B cells and some epithelial and fibroblastic cells and promote their activation following the binding of ligands to their cognate receptor [44,45].

The PRR-targeting adjuvants licensed for human use are 3-O-desacyl-4′-monophosphoryl lipid A (MPLA) and cytosine phosphoguanosine (CpG) 1018 [43]. MPLA is a purified form of lipopolysaccharides (LPS) from the bacteria *Salmonella minnesota* and activates the TLR4, a type of PRR. It was first used in combination with alum under the proprietary name AS04 (GSK) in HPV and HBV subunit vaccines. The combination of MPLA and alum induces higher antibody titers compared to alum alone in both vaccines while increasing the breadth of protection against multiple strains of HPV [41].

Bacterial proteins with the ability to activate membrane-expressed TLR2, TLR4, or TLR5 also show great potential as immune-enhancing components in vaccines. Flagellin was the first known agonist for TLR5. Its adjuvant properties were first evaluated thirty years ago, and it has been extensively used in the context of vaccine research in pre-clinical and clinical settings [46]. Flagellin can promote the maturation of APCs, inducing the secretion of pro-inflammatory cytokines, and increasing the level of antibodies directed toward an antigen when injected in a co-mixture, as a fusion protein, or incorporated into nanoparticles. Since TLR5 is highly expressed in the airway epithelial and immune cells, flagellin shows potential as a mucosal adjuvant for intranasal or oral vaccines [46]. A recent study has identified the protein P97c from Mycoplasma hyopneumonia as a novel TLR5 agonist capable of inducing concentration-dependent cytokine production in HEK-blue mTLR5 (mouse TLR5) cells. Furthermore, its efficacy as an adjuvant was demonstrated by conjugating the ectodomain matrix 2 protein (M2e) of influenza A, leading to higher M2e-specific antibody titer following mice immunization [47]. Another membrane PRR, TLR2 is more promiscuous than TLR5 and recognizes lipoproteins and a wide variety of hydrophobic proteins [48,49,50]. It forms a heterodimer with TLR1 or TLR6. A multitude of bacterial proteins have TLR2 agonist activity, mostly cell surface-expressed proteins like porins or outer membrane proteins (OMPs). Bacterial proteins with TLR2 binding properties have been shown to promote leukocyte recruitment, induce the production of pro-inflammatory cytokines, the maturation of APCs and the production of antigen-specific antibodies and cellular response. While most bacterial proteins with TLR2-activation capacities induce a Th1-skewed immune response, some have reported a balanced Th1-Th17 immune response, mainly in the lungs [51,52]. Interestingly, some TLR2-targeting proteins can activate TLR4 [51]. While LPS and its derivatives are the most characterized TLR4 agonists, it is becoming increasingly clear that a myriad of bacterial proteins have the same ability. These proteins have been shown to promote the maturation and migration of DC to lymph nodes, the production of pro-inflammatory cytokines and a robust B and T cell activation. Similar to TLR2 signaling by bacterial proteins, TLR4 agonists induce a Th1-Th17 immune response [51]. Moreover, bacterial components that activate the immune system in a TLR-independent manner have also been studied for their potential use as adjuvants, for example, the heat-labile enterotoxin (LT) and the cholera toxin (CT). Early studies on their underlying mechanism point to the accumulation of cAMP as adjuvant effects and the activation of the Nod-like receptor Pyrin 3 (NLRP3) [53,54]. The LT protein induces strong Th17 responses and increases sIgA titers when used as a mucosal adjuvant. However, adverse side effects were reported in clinical studies and linked the adjuvant to Bell’s Palsy symptoms, including facial paralysis, in early 2000 [55]. This prompted research on genetically modified LT with reduced toxicity in humans for adjuvant uses (reviewed in [56]).

### 4.3. Conjugation to Nanoscale Antigen Delivery Systems

Since the immune system has evolved to recognize pathogens under their particulate forms, mimicking this structure in subunit vaccines could potentially increase the immune response generated following administration. Accordingly, the conjugation of antigens with particles increases their immunogenicity and physical and metabolic stability, while limiting potential toxicity. Antigens under particulate forms enhance uptake by APCs, lymph node trafficking and persistence, which leads to a stronger cellular and humoral immune responses being directed against the antigen [28,42,57,58,59,60,61,62,63,64,65].

The biodistribution of antigens following administration is critical for vaccine effectiveness. Particle size is a parameter that greatly affects the pharmacokinetics of vaccines. Nanoparticles under 5 nm readily diffuse in the blood where they circulate systemically, while the ones between 10 nm and 100 nm drain in the lymphoid system toward LN [61,66,67,68,69,70,71]. Particles over 100–200 nm are mainly trapped in the extracellular matrix at the site of injection and must be brought to LN via APCs [42,72,73]. Shape, rigidity, and surface chemistry are also important factors influencing the biodistribution of particles [42,57,64]. Limited circulation is a clear advantage of nanoparticles because the systemic dissemination of small molecules limits the efficacy of vaccination while potentially increasing the side effects [28,74]. Once in the LN, particles have been shown to persist for a longer period compared to smaller molecules. While nanoparticles smaller than 15 nm are rapidly found in the follicles of the LN following immunization, they are also rapidly eliminated. On the contrary, particles of 50–100 nm tend to take more time to reach the follicles but can persist for a few weeks when immobilized on FDCs [28,75]. It is interesting to note that this phenomenon seems to be dependent on a complement, which opsonizes the particles and promotes their retention by the FDCs via their complementary receptors [75,76]. The key advantages of conjugating antigenic materials on nanoparticles for subunit vaccines are summarized in Figure 4.

Nanoparticle-associated antigens can also be brought to the LN from the site of injection by APCs. Particles that are over 20 nm are more efficiently internalized by DC and macrophages compared to soluble antigens [66,77,78,79]. The repetitive nature of nanoparticles also enhances internalization by APCs via multiple mechanisms that are not fully understood. Amongst others, the binding of natural antibodies to repetitive patterns triggers the recruitment of the complement which, in turn, interacts with Fc receptors (FcR) and promotes internalization of the opsonized material by APCs [57].

To enhance the presentation of antigen-derived peptides on MHC molecules, APCs must be activated. The co-delivery of adjuvant and antigens by particles promotes internalization, antigen processing, maturation of the APC and presentation of the antigen-derived peptide at the surface of the cell. Danger signals also promote the cross-presentation of exogenous peptides on MHC-I molecules by DCs [80,81]. Since the inflammation generated by activated immune cells is usually the source of vaccine side effects, targeting adjuvants to antigen-experienced cells offers the possibility to decrease systemic toxicity while promoting vaccine effectiveness [28]. The co-delivery of antigen and adjuvant also promotes B cell engagement and enhances GC formation and the humoral immune response generated toward the antigen [82].

Moreover, nanoparticles can present a high density of antigens at their surface. This allows for a repetitive display, similar to virus particles [83]. Multivalent antigens efficiently induce BCR cross-linking, which greatly enhances uptake and presentation by high- and low-affinity B cells, compared to the soluble protein, which is mostly uptaken by B cells with high-affinity BCR. The engagement of low-affinity B cells could be useful in the generation of broadly neutralizing antibodies (Nabs) [58]. It was also shown that multivalent antigens can enhance T cell activation by promoting presentation by B cells while increasing antigen-specific antibodies by up to 10-fold [84,85,86].

In summary, the display of antigens on nanoparticles offers many advantages over soluble antigens (Figure 4). Notably, nanoparticles are preferentially uptaken by APCs, and efficiently drain to lymph nodes, where they are retained for a longer period and offer limited systemic toxicity. Furthermore, the use of nanoparticles favors the co-delivery of antigen and adjuvant to the same cell, which enhances adjuvant effects and limits off-target effects. The conjugation of antigen on nanoparticles stabilizes them and allows for the display of antigens under “locked” conformation (i.e., pre-fusion stabilized GP). Moreover, antigen multivalency on nanoparticles can facilitate BCR cross-linking and antibody production [87]. It has also been demonstrated that the multivalent display of an antigen on nanoparticles not only significantly improves their immunogenicity, but also induces potent immune responses at relatively low immunogen dose when compared to animals immunized with the soluble antigen alone [88,89].

Recent developments have made it possible to design nanoparticles with distinctive physicochemical characteristics. One may construct nanoparticles with specific biological features by tuning and controlling factors including size, shape, solubility, surface chemistry, and hydrophilicity. These characteristics suggest that nanoparticles are promising immune cell stimulators and antigen carriers for immunization. These nanoscale materials can be conceived de novo or derived from living organisms. A wide range of particles, including inorganic and polymeric nanoparticles, virus-like particles (VLPs), liposomes, and self-assembling protein-based nanoparticles, have been evaluated as antigen carriers [71]. Since many bacterial proteins can self-assemble into nanoparticles of defined structure, they are interesting candidates for the generation of nanoparticle-based subunit vaccines with enhanced immunogenicity. They self-assemble into highly symmetric, stable nanoparticles with diameters of 10–150 nm [71,83], which is an ideal size range ideal for interacting with different immune cells [83]. Since they can be used as nanoplatforms for the organized display of specific immunogen, these nanoparticles are of particular interest in the context of vaccine design because they can mimic the repetitive surface architecture of most naturally occurring pathogens.

## 5. Self-Assembling Bacterial Proteins as Nanoscaffolds for Antigen Delivery in Subunit Vaccines

In recent decades, bacterial protein-based nanoparticles have been evaluated in numerous subunit vaccine formulations [57,90,91,92,93]. The unique particulate nature and repetitive subunit organization of these assemblies make them ideal candidates for antigen display, which, in turn, would provide a robust antigen-specific immune response. These self-assembling bacterial protein-based nanoparticles for the delivery of antigenic determinants are summarized in detail in Table 2 and Figure 5.

### 5.1. Ferritin

Almost all living species, including bacteria, fungi, plants, and mammals, expressed the protein ferritin [134,135]. Its main physiological function is to store iron in an insoluble, non-toxic form while making it intracellularly accessible by converting it to a soluble form [136], playing a crucial role in iron homeostasis. It also protects against free-iron-related toxicity, such as the production of reactive oxygen species, which can damage cellular machinery and cause cell death [137]. Structurally, ferritin nanoparticles have a hollow core with inner and outer diameters of 8 and 12 nm that may internalize up to 4500 iron atoms in the form of ferric oxyhydroxide [138], as well as varying quantities of phosphate [139]. Each ferritin particle consists of 24 identical or homologous subunits that self-assemble in an octahedral (432) symmetry [135]. Ferritin has lately emerged as a promising antigen-displaying platform [140]. In addition to its ability to self-assemble, the ferritin protein complex exhibits remarkable thermal and pH stability, biocompatibility, biodegradability and is significantly cost-effective for large-scale production, hollow cavity with reversible assembly/disassembly, and amenability to surface conjugation via chemical or genetic approaches [141]. Ferritin has been employed in nanobiotechnology for drug delivery, biomimetic synthesis, bioimaging, and cell targeting [135,141,142,143,144,145].

*Helicobacter pylori*, *E. coli* and *Pyrococcus furiosus* 24-mer non-heam self-assembling ferritin have been most widely used for the development of bacterial protein-based nanoparticle subunit vaccines (Figure 5A), notably against SARS-CoV-2 (using the spike (S protein), RBD and heptad repeat (HR)) [89,94,95,96,110,114], Middle East respiratory syndrome-coronavirus (MERS-CoV) [111], IAV [97], IAV subtype H5N1 [98], Ebola virus (EboV) [116], RSV [100], hepatitis C virus (HCV) [101], HIV [102,103,104,105,112], rotavirus A [106], porcine reproductive and respiratory syndrome virus (PRRSV) [107], foot-and-mouth disease virus (FMDV) [108], Epstein-Barr virus (EBV) [99], classical swine fever virus (CFSV) [109], HBV [113] and HPV [115]. These self-assembling ferritins have been produced in a variety of hosts including bacteria (E. coli BL21 (DE3)), insects (Sf9, High-Five and Drosophila S2 cells) and mammalian cells (Expi293F, ExpiCHO, FreeStyle™ 293-F, FreeStyle™ HEK293F, HEK293S and FreeStyle™ CHO-S). The antigens were fused to the ferritin using either a genetic fusion, chemical modification, or the SpyTag/SpyCather system. Upon self-assembling, ferritin nanoparticle-based vaccine candidates yielded nanoparticles with 20–50 nm in diameter. The advantage of self-assembling in 24-mer is that 24 copies of antigen/epitope can be genetically or covalently conjugated. These self-assembling vaccine candidates have been used in various model animals, such as mice, rabbits, pigs, non-human primates and many others, to study humoral and cellular immunity.

Using ferritin nanoparticles, Ma et al. (2020) [95] exposed that ferritin-based nanoparticle vaccines induce potent Nabs and cellular immune responses against SARS-CoV-2. Furthermore, the vaccination of transgenic hACE2 mice with RBD and/or RBD-HR nanoparticles demonstrated a significantly reduced viral load and elevated protection following SARS-CoV-2 infection. Moreover, nanoparticles also assisted in inducing Nabs and cellular immune responses against other coronaviruses. Additionally, the authors have exhibited that the use of these nanoparticles also induced Nabs, as well as T cell responses in Rhesus Macaques, which persisted for more than three months. Of note, they also assessed that the nanoparticles did not induce ADE. In another study, Wang et al. (2021) [114] showed that a ferritin nanoparticle-based SARS-CoV-2 RBD vaccine generated an effective antibody response in mice that sustained for at least 7 months after inoculation. Moreover, they also reported that, upon antigen exposure, a large proportion of MBCs were preserved and considerably recalled and induced persistent antibody response. They have also exhibited that ferritin nanoparticle vaccine with preS1 domain of HBV elicited a therapeutic antibody response against chronic hepatitis B in a mouse model [113]. In nonhuman primates, Joyce et al. (2021) [96] designed and tested the SARS-CoV-2 spike ferritin nanoparticle (SpFN) vaccine, which produced a Th1-biased CD4 T cell response and Nabs against wild-type SARS-CoV-2, variants, as well as SARS-CoV-1. Following a high-dose SARS-CoV-2 respiratory challenge, these powerful humoral and cell-mediated immune responses resulted in the fast clearance of replicating virus in the upper and lower airways, as well as the lung parenchyma of nonhuman primates. In another example, an SpFN nanoparticle vaccine candidate was paired with two distinct adjuvants, Alhydrogel^®^ or Army Liposome Formulation containing QS-21 (ALFQ) [94]. They exhibited that SpFN-ALFQ efficiently activates innate immune cells and improves SARS-CoV-2-specific long-lasting adaptive immune T cell responses. Innate immune cell activation was linked to robust antigen-specific polyfunctional cytokine responses and cytolytic activity. The role of the adjuvant ALFQ, together with ferritin nanoparticles, was responsible for directing the interaction of innate and adaptive immune responses.

For reliable and repeatable manufacturing of nanoparticle-based vaccines, monomeric antigen folding and subsequent assembly into highly ordered structures are critical. Despite substantial breakthroughs in silico design and structure-based assembly, most engineered nanoparticle-based vaccines are resistant to soluble expression and fail to assemble as intended, posing severe manufacturing problems in nanoparticle-based vaccine development. Therefore, the RNA-interaction domain (RID) was used by Kim et al. (2018) [111] as a reliable protein-folding vehicle to assemble NPs in bacterial hosts. They genetically fused the MERS-CoV RBD with Helicobacter pylori ferritin and RID to produce nanoparticles in a soluble form in Escherichia coli. The results exhibited that MERS-CoV RBD binding to the cellular receptor hDPP4 was efficiently inhibited by mice sera following vaccination. In conclusion, their findings demonstrated that RID regulates the antigen folding pathway’s overall kinetic network in favor of improved nanoparticles assembling into highly regular and immunologically relevant conformations. Moreover, the concentration of the Fe^2+^, salt, and fusion linker all played a role in in vitro assembly and stability of these nanoparticles.

Kanekiyo et al. (2013) [97] genetically linked the haemagglutinin (HA) of IAV (H1N1) to ferritin nanoparticles. Immunization with this influenza nanoparticle vaccine resulted in haemagglutination inhibition antibody titers that were more than tenfold greater than those obtained with the approved inactivated vaccine. It also evoked Nabs to the stem and the receptor-binding site on the head of HA, two highly conserved sites that are universal vaccine targets. As a result, these self-assembling nanoparticles induced a broader and more powerful protection than standard influenza vaccinations. As mentioned above, the subdominant stem region of HA is highly conserved and identified by antibodies that can bind different HA subtypes. Therefore, Yassine et al. (2015) [98] developed a ferritin nanoparticle-based HA H1 stem-only GP vaccine candidate using a structural-based design approach. Despite the lack of detectable H5N1 neutralizing activity in vitro, the vaccination of mice and ferrets with ferritin nanoparticle-based stem-only HA resulted in widely cross-reactive antibodies that totally protected mice and partially protected ferrets against deadly heterosubtypic H5N1 IAV challenge. Moreover, the passive transfer of antibodies from immunized mice to naïve mice protected them from the H5N1 challenge, demonstrating that the vaccine-elicited HA stem-specific antibodies can protect against a variety of group 1 influenza viruses.

To enhance antigen presentation and target antibody responses to important epitopes of the F protein of RSV, Swanson et al. (2020) [100] used a structure-based rational design and fused a stabilized pre-F protein to ferritin nanoparticles (pre-F-NP) (Figure 5A). They also concealed non-neutralizing epitopes with glycans. The multimeric pre-F-NP induced long-lasting specific Nabs in nonhuman primates and mice over 150 days.

Chen et al. (2020) [108] investigated ferritin nanoparticles that were recombinantly linked to VP1 and G-H loop subunits of FMDV, an acute, febrile, and highly contagious infectious disease common in cloven-hoofed animals. Their findings indicated that ferritin nanoparticles containing recombinant proteins were immunogenic. Recombinant FMDV subunit vaccinations boosted FMDV-specific IgG (IgG1 and IgG2a) antibody titers, as well as IL-4 and IFN-ɣ production. The ferritin nanoparticles also provided partial protection in mice.

Li et al. (2019) [106] proposed an experimental rotavirus vaccine transgenically expressed in the milk of mice. It was tested for immunological protection in a mouse model in order to create a rotavirus vaccine that would be suited to both mammary-gland-based manufacturing and milk-based administration. Their findings implied that the recombinant VP6–ferritin nanoparticle vaccine can effectively prevent the mortality and malnutrition caused by rotavirus infection in pups, making it a promising candidate vaccination for rotavirus.

Trimers of the HIV-1 envelope (Env) are typically poorly immunogenic. Therefore, Sliepen et al. (2021) [105] compared the efficacy of several adjuvants (squalene emulsion, ISCOMATRIX, GLA-LSQ, and MPLA liposomes) to promote Nab responses in rabbits using the clinically relevant ConM SOSIP.v7 trimer fused to ferritin nanoparticles. When the ferritin nanoparticles were delivered with ISCOMATRIX, stronger Nab responses were evoked. In conclusion, they exhibited that the ferritin nanoparticle’s immune response could be enhanced with the combination of adjuvant, but the nature of the antigens and nanoparticles must be taken into consideration. In other studies, they reported that HIV-1 Env GP trimers bearing ferritin nanoparticles were significantly more immunogenic than trimers in both mice and New Zealand white rabbits [103] and elicited a strong Nabs response against the autologous virus in New Zealand white rabbits and Rhesus macaques [104].

Zhao et al. (2021) [109] reported a complete vaccination method for displaying the E2 GP of the CSFV on the surface of self-assembling ferritin nanocages. In vivo data showed that rabbits inoculated with ferritin nanoparticles triggered both humoral and cellular immunity as indicated by Nab titers and expression of IL-4 and IFN-ɣ.

In an in vitro study, He et al. (2015) [101] developed and characterized epitope vaccine antigens targeting the antigenic locations of HCV envelope GP E1 (residues 314–324) and E2 (residues 412–423). They then used a “multivalent scaffolding” strategy to improve antibody binding avidity by displaying 24 copies of an epitope scaffold on a ferritin self-assembling nanoparticle. Their research shows the value of a multi-scale scaffolding technique in epitope vaccine development and identifies prospective HCV immunogens for in vivo testing. The nanoparticles had delayed off-kinetics for both antigenic sites, indicating a substantial avidity impact owing to multivalent antibody binding.

Due to its capacity to elicit responses against a greater variety of distinct HPV strains, the HPV minor capsid protein L2 has been studied as a possible antigen candidate substitute for major capsid protein L1. In this regard, the ferritin-L2 antigen nanoparticles caused a broadly Nab response in guinea pigs and mice that covered 14 oncogenic and two non-oncogenic HPV strains. The immune response lasted for at least one year and provided protection in a cervico-vaginal mouse model of HPV infection [115].

In contrast to inactivated PRRSV, Ma et al. (2021) [107] showed that a modified envelope glycoprotein 5 (GP5)-ferritin nanoparticle vaccine produced greater serum antibody titers in pigs. Additionally, the nanoparticle vaccine boosted a Th1-dominant cellular immune response and enhanced specific T lymphocyte immune responses. In comparison to unvaccinated pigs, GP5-ferritin-vaccinated pigs had significantly lower mean rectal temperatures, respiratory scores, viremia, and scores for both macroscopic and microscopic lung lesions after the challenge. These findings demonstrated the potential of ferritin subunit vaccines to induce protective immune responses and serve as vaccine candidates.

### 5.2. Lumazine Synthase

Lumazine synthase (LS, EC 2.5.1.78) is an enzyme that catalyzes the penultimate step in the biosynthesis of riboflavin, widely known as vitamin B2 [146]. LS from *Bacillus subtilis*, hyperthermophilic bacterium *Aquifex aeolicus* and a variety of other bacteria and archaea produce icosahedral capsids with triangulation (Figure 5B). The icosahedral compound has a triangulation number (T) = 1 [147,148,149,150]. The capsids have an outside diameter of roughly 15–16 nm and are made up of 12 pentameric units, totaling 60 identical subunits linked by symmetry axes of twofold, threefold, and fivefold of ~960,000 Daltons. In an alkaline medium (pH > 8), *Bacillus subtilis’* LS transforms from T = 1 state to a T = 3 state, which is composed of 180 identical subunits with a diameter of roughly 29 nm [149]. This transformation is caused by the loss of a phosphate ion per monomer, which stabilizes the T = 1 state [146,149]. LS’s symmetric nanoparticle carriers have been shown to display a structurally ordered array of immunogens, which are summarized here.

In this regard, using the SpyTag/SpyCatcher system, Zhang et al. (2020) [89] created a modular 60-subunit *Aquifex aeolicus* LS-based self-assembling nanoparticle (in parallel with the 24-subunit Helicobacter pylori ferritin) platform that enables the plug-and-play display of trimeric viral GP on nanoparticle surfaces (Figure 5B). This technique was tested using three viral trimeric GP that were pre-fusion (preF)-stabilized: RSV fusion (RSV F) GP, human parainfluenza virus type 3 fusion (PIV3 F) GP, and SARS-CoV-2 S GP. The higher antigenicity of the apical epitopes of trimeric viral GP attached to LS nanoparticles resulted in improved immunogenicity, especially at lower doses. The vaccination of mice with 0.08 μg of SARS-CoV-2 spike-LS nanoparticle induced identical neutralizing reactions as 2.0 μg of the spike, which was 25-fold greater in terms of weight-per-weight. Aebischer et al. (2021) [120] also established a durable and adaptable self-assembling multimeric protein scaffold particle (MPSP) vaccination platform using LS from Aquifex aeolicus. They used the SpyTag/SpyCatcher-mediated plug-and-display on the pre-assembled particles of LS-MPSPs to display two model antigens of Schmallenberg virus (SBV) and studied their efficacy in mouse and cattle models. In both models, they showed that the nanoparticles improved immunogenicity and vaccine efficacy. For example, a single dose of this vaccine protected roughly 80% of mice and gave cattle an almost sterile immunity against an otherwise deadly dosage of SBV.

Furthermore, Tokatlian et al. (2019) [112] evaluated the fate of two different extensively glycosylated HIV antigens, gp120 and gp140. To generate protein nanoparticles, the gp120 antigen was fused to a bacterial protein LS, and, on the other hand, archaeal ferritin was used to fuse the gp140 antigen. Unlike monomeric antigens, multivalent glycosylated antigens displayed on nanoparticles stimulate mannose-binding lectin-mediated innate immune recognition in vivo, resulting in fast complement-dependent transport to FDCs and their subsequent accumulation in GCs. This focused trafficking was linked to improved antibody responses, suggesting that immunogen glycosylation may be a significant design requirement for future nanoparticulate vaccines. These findings are particularly intriguing in the context of HIV vaccine development, where the thick envelope “glycan shield” is frequently seen as a barrier to eliciting effective antibody responses. Designing immunogens that can trigger broadly Nabs that bind to the HIV-1 viral Env GP is one of the challenging aspects of HIV-1 vaccine research. LS nanoparticles, when multimerized with a rationally designed epitope (HIV-1 gp-120), showed that formulated nanoparticles activate germline and mature VRC01-class B cells [118] and could prime a broadly Nabs response to HIV-1 [117].

### 5.3. Encapsulin

Encapsulin is a nanocarrier particle that contains virus capsid-like nanocompartments found in a variety of bacteria and archaea [93,151,152,153,154]. It has been demonstrated that these nanocompartments may store iron and protect bacteria from oxidative stress [155]. Structurally, encapsulin nanocompartments differ from organism to organism. For example, *Rhodococcus erytropolis* [156], *Mycobacterium tuberculosis* [157], and *Thermotoga maritima* [153] encapsulin are made up of 60 identical subunits that form T = 1 icosahedral capsid-like particles with a diameter of 20–24 nm. On the other hand, *Myxococcus xanthus* [155] and *Pyrococcus furiosus* [151] encapsulin nanocompartments comprise of 180 protein subunits in a T = 3 icosahedral particles with a diameter of from 30 to 32 nm. Encapsulin nanocompartments package functional enzymes such as ferritin-like proteins and Dyp-type peroxidases in bacteria [93,153,155,157]. A specific C-terminus sequence of these encapsulated proteins causes them to bind to the internal surface of the encapsulin [153]. Taking advantage of this, many non-native cargo proteins (i.e., antigens) were packaged using the unique cargo-loading ability of the encapsulin nanocompartments (Figure 5C) [158,159].

Lagoutte et al. (2018) [93] demonstrated the ability of *Thermotoga maritima’s* encapsulin nanoparticles for simultaneous surface display of M2e epitope of IAV and packaging of a green fluorescent protein (GFP) reporter into the internal cavity. In this study, the researchers successfully demonstrated that the engineered encapsulin nanoparticles facilitated both surface display and packaging properties. This surface engineering of encapsulin nanoparticles also enhanced the cargo-loading capacity of the heterologous reporter protein. Furthermore, the immunogenicity study in mice revealed strong antibody responses against both the surface epitope and the loaded cargo protein without compromising the booster immune response to the targeted epitope. Their study represents the enormous potential of encapsulin nanoparticles for rational vaccine design and antigen delivery. Kanekiyo et al. (2015) [99] have developed an EBV vaccine candidates based on encapsulin and ferritin self-assembling nanoparticles (Figure 5C). These platforms elicited potent and long-lasting virus-Nabs in mice and nonhuman primates that target the receptor-binding site on the viral envelope protein gp350. More specifically, they blocked gp350’s CR2-binding site and exhibited enhanced vaccine-induced protection in a mouse model. Of note, the ferritin-based nanoparticles provided an 80% survival rate versus 20% for the encapsulin-based nanoparticle in the mice model.

### 5.4. sHSP and P22

The small heat shock protein 16.5 (sHsp) from *Methanocaldococcus jannaschii* (a hyperthermophilic archaeon) [122,160,161] is composed of 24 repeating subunits. Due to the high symmetry and quaternary structure, these subunits self-assemble to form an empty cage-like shape, similar to that of a viral capsid (Figure 5D) [123,132,162]. Flenniken et al. (2003) and Abedin et al. (2009) [132,163] previously exhibited that sHsp may be genetically modified to contain cysteine residues, therefore enabling attachment sites for bioconjugation [132,163,164], which has been used for the display of a foreign protein [165,166]. On the other hand, P22 is a bacteriophage capsid that infects *Salmonella typhimurium* in the presence of intact tail fibers [167,168]. The P22 employed for vaccine developments lacks both genetic material and tail fibers, leaving just the non-infectious empty viral capsid (Figure 5E). P22 forms an icosahedral procapsid (58 nm in diameter) from 415 copies of the 46.6 kDa coat protein and roughly 300 copies of the 33.6 kDa scaffolding protein [122]. Richert et al. (2012) [122] used sHsp and P22 phage-derived VLPs as immunomodulatory platforms in both nonspecific pre-priming scenarios and as the delivery of particular antigens to the lung. Specifically, they exhibited that ovalbumin (OVA) could be chemically fused to the outside of an sHsp cage. When naive mice were given OVA–sHsp intranasally, the immune response to OVA was expedited and strengthened and OVA-specific IgG1 responses were observed 5 days after a single dose, demonstrating its potential for vaccine delivery platform. Furthermore, the research group showed a strong mucosal sIgA titer alongside GCs for B and Tfh cell accumulation. As a result, they demonstrated that sHsp and P22 VLPs may be employed as both immunomodulatory agents and antigen carriers, allowing for local immunization of the lower respiratory tract against pathogens with a single dose. In another study, Wiley et al. (2009) [123] generated protein cage nanoparticles (PCNs) derived from the sHsp 16.5, which comprised 24 identical subunits that spontaneously self-assemble into hollow, spherical protein cages that were 12 nm in diameter. They showed that mice pre-treated with PCN, independent of any viral antigens, were protected from both sub-lethal and fatal dosages of two distinct IAV, a mouse-adapted SARS-coronavirus, or mouse pneumovirus. Treatment with PCN markedly improved viral clearance, expedited the induction of viral-specific antibody production, reduced morbidity and lung damage and greatly increased survival.

To study the response of P22 VLP to IAV variability, Sharma et al. (2020) [125] used a mouse model to assess the immunogenic potential and protective effects of P22 VLPs conjugated with multiple copies of the globular head domain of the HA protein from the PR8 strain of IAV. The HA globular head was coupled to preassembled P22 VLPs via a covalent attachment technique (SpyTag/SpyCatcher). Mice immunized with this P22-HA head combination were completely protected from morbidity and mortality after infection with a homologous IAV strain. The authors also exhibited that P22 VLPs may be quickly modified in a modular way to design potent vaccine(s) that can produce protective immunity without the use of additional adjuvants. Similarly, Patterson et al. (2013) [124] developed a vaccine strategy where the antigenically conserved nucleoprotein from influenza was fused on the interior of P22-derived VLP. They reported that the P22-derived vaccine protected mice against many strains of IAV (H1N1 and H3N2) in a nucleoprotein-specific CD8+ T cell-dependent manner. Their findings also highlighted the P22 system’s ability to integrate large antigenic proteins, which is frequently a limiting element in other VLP systems.

### 5.5. BP26

Through the self-assembly of 16 monomeric proteins, BP26, an OMP of the zoonotic pathogen *Brucella abortus*, creates a barrel-like shape with a hollow core (Protein Data Bank ID: 4HVZ) (Figure 5F) [126,169]. As BP26 is a common immunodominant antigen in *Brucella* bacteria, its barrel-like nanostructure exhibits potent adjuvant efficacy [170]. As a result, it has been hypothesized that engineered BP26 monomers containing an antigen could self-assemble into antigen-displaying BP26-based nanobarrels. These nanobarrels have been shown to improve recognition by BCRs, resulting in increased antigen-specific antibody production [59,62,171]. Kang et al. (2021) [126] created a universal IAV vaccine platform that is cross-protective using the nanobarrel self-assembly of BP26 to build a protein nanoarchitecture displaying the extracellular domain from M2e of IAV (BP26-M2e) (Figure 5F). Mice immunized with BP26-M2e nanobarrel vaccines produced high levels of anti-M2e antibodies with antibody-dependant cell cytotoxicity capacity. Furthermore, their platform induced T-cell responses and effectively protected mice from IAV infection-associated death, even without the use of a conventional adjuvant.

### 5.6. Flagellin

The flagellum of bacteria is made up of the protein flagellin. Due to its capacity to trigger elements of the innate immune system, flagellin has shown considerable promise as a vaccine adjuvant [172,173,174,175]. This protein has attracted interest for its use as an adjuvant in vaccine formulations as a powerful TLR5 agonist. TLR5 activation increases the synthesis of IL-6 through the MyD88-dependent pathway [172]. Flagellin can also activate the NAIP5 and NLRC4 NOD-like receptors (NLRs), which, in turn, causes the NLRC4 inflammasome to assemble and activates caspase-1, resulting in proinflammatory signals [176,177,178]. Although flagellin is capable of self-assembling into linear filaments (nanotubes), the majority of research to date has only employed the protein’s monomeric form as an adjuvant [128]. However, monomeric flagellin-based vaccines have been proven to cause visible and long-lasting side effects, such as discomfort at the injection site, exhaustion, and muscular pains due to the sustained activation of TLR5, which has compromised their use in clinical trials [127,179]. In contrast, the use of nanotubes as an adjuvant cannot be effectively internalized by APCs or passively drained to the lymph nodes due to their aspect ratio [57]. Additionally, studies have demonstrated that when compared to their monomeric counterpart, flagellin nanotubes cause a significantly lower activation of TLR5 and reduced stimulation of the innate immunity [180,181]. To avoid using µm-long nanotubes as innately immunostimulatory antigen delivery platforms, it may be possible to manipulate flagellin’s ability to self-assemble into lower-aspect-ratio supramolecular nanostructures [127].

Flagellins have from two to four domains, with D0 and D1 playing a role in the TLR5 interaction [182,183]. On the other hand, the D2 and D3 domains, display considerable sequence variation between species and are involved in flagellated pathogen immune evasion [184]. Evidence shows that the D0 and D1 domains are involved in the self-assembly aspect of this protein [185]. Côté-Cyr et al. (2022) [179] fused an IAV M2e epitope to self-assembling flagellins Hag from *Bacillus subtilis* (having only D0 and D1 domains) and FljB from *S. Typhimurium* (containing D0, D1, D2 and D3 domains) [182] to investigate their immunostimulatory and pro-inflammatory characteristics. Both flagellins activated TLR5, but FljB was 2.5 times more potent than Hag. However, mice inoculated with FljB or Hag elicited a strong M2e-specific antibody response, with Hag showing reduced production of pro-inflammatory markers and weight loss. Therefore, the study showed that flagellin Hag was a powerful immunoadjuvant with minimal side effects. In another study, Côté-Cyr et al. (2022) [127] also described a method for controlling the self-assembly of the *Bacillus subtilis* flagellin protein Hag into reduced aspect ratio nanoparticles by preventing the non-covalent contacts that cause the protein to elongate into nanotubes. They exhibited that adding three repetitions of the M2e antigenic sequence from the IAV to the C-terminus of flagellin prevented filament elongation and led to low-aspect-ratio ring-like nanostructures during salting-out-induced self-assembly (Figure 5G). APCs successfully absorbed flagellin-ring-like nanostructures, which, in turn, triggered TLR5 response in vitro as well as innate and adaptive immune responses. In summary, the authors reported that these nanostructures have the potential to act as antigen carriers because they are intrinsically immunostimulatory. The intranasal vaccination of mice with these nanostructures led to the potentiation of the antigen-specific antibody response and protection against a deadly IAV infection.

### 5.7. Other Self-Assembling Proteins with Potential Usage in Subunit Vaccines

A new development in bacterial protein-based self-assembling nanoparticles has been the emergence of computationally designed protein nanoparticles as a robust and versatile platform for multivalent antigen presentation [129,186,187,188,189,190,191]. In preclinical studies, vaccine candidates based on designed protein nanoparticles have significantly improved the potency or breadth of antibody responses against numerous antigens, including prefusion RSV F [129], Env from HIV-1 [192], HA from IAV [193], and *P. falciparum* cysteine-rich protective antigen (CyRPA) [194], relative to either soluble antigen or commercial vaccine comparators. Among these platforms, I3-01 (derived from *Thermotoga maritima*) [187], E2p (derived from *Geobacillus stearothermophilus*) and I53-50 [186] are the most extensively used, computationally designed, self-assembling subunit vaccines.

I3-01 is derived from *Thermotoga maritima’s* 2-Keto-3-deoxy-6-phosphogluconate (KDPG) aldolase, which is pyruvate aldolase central to the Entner–Doudoroff glycolytic pathway [119,187,195,196]. In this metabolic pathway, glucose and galactose are converted to the corresponding 3-deoxy-6-phosphohexulosonate and then cleaved enzymatically to pyruvate and d-glyceraldehyde-3-phosphate [195,196]. The KDPG aldolase protein forms a capsule-like protein that resembles a virus capsid. Therefore, the KDPG structure has been studied and computationally modified to generate I3-01 (24-kDa), which assembles into a 25 nm dodecahedron (Figure 5H) [119]. This ability to design proteins that self-assemble into precisely specified, highly ordered icosahedral structures has opened the door to a new generation of protein ‘containers’ that could exhibit properties custom-made for various vaccine and drug-delivery applications.

Furthermore, *Geobacillus stearothermophilus* E2p (26-kDa), which self-assembles into a pentagonal dodecahedral scaffold of 27 nm, is an icosahedral scaffold formed by the acyltransferase component (E2 protein) of the multienzyme pyruvate dehydrogenase complex (PDH) [197]. E2p has been reported for its ability to display peptides in a highly immunogenic form [198,199]. The core C-terminal catalytic domain of E2p self-assembles into trimers, which, in turn, aggregate to generate a 60-chain core with an icosahedral symmetry [200,201,202]. Moreover, this 60-meric icosahedral structure can be regenerated with high efficiency from denaturing conditions in vitro, without the need for chaperonins [203,204]. The robustness of this peptide-based VLP rendered it an attractive macromolecular scaffold for the presentation of exogenous molecules on its surface [198,199,200] and for molecular encapsulation in its cavity [205,206]. Efficient refolding of E2p to the 60-mer is also possible, with foreign peptides replacing the natural peripheral domains, as N-terminal fusions to the core domain. Thus, a suitably engineered E2p core (E2DISP) can display 60 copies of heterologous peptides on the surface of a high-molecular-mass scaffold [198,199,200], which could be exploited for vaccine design (Figure 5H).

I53-50 is another computationally designed, two-component protein complex comprising 20 trimeric ‘‘A’’ components and 12 pentameric ‘‘B’’ components for a total of 120 subunits with icosahedral symmetry that is 28 nm broad (Figure 5I) [131,186]. I53 stands for icosahedral assembly constructed from pentamers and trimers. One of the major advantages of using I53-50 is that these nanoparticles could be easily constructed in vitro by simply combining I53-50A and I53-50B that have been produced and purified separately, a characteristic that has aided in its usage as a platform for multivalent antigen presentation [131]. Altogether, such robustly designed nanostructures of I3-01, I53-50 and E2p were studied to have considerable utility for targeted vaccine design, which are discussed and summarized here.

He et al. (2021) [116] demonstrated a rationally designed GPΔmuc trimer to 60-mer E2p and I3-01 (as well as 24-mer ferritin) protein nanoparticles (Figure 5H). These nanoparticles were re-engineered in such a way that a dimeric locking domain (LD) is fused to the C terminus of a nanoparticle subunit and a Th cell epitope. These GP trimers and nanoparticles were observed to induce cross-EBV Nabs in mice and rabbits. In another study, He et al. (2016) [102] exhibited that trimeric HIV-1 antigens gp120 and gp140 trimer based on 60-meric E2p with 20 spikes (in parallel with ferritin) resemble VLPs. They found that gp120 and gp140 nanoparticles provide a strong activation of B cells with cognate VRC01 receptors in vitro. Kang et al. (2021) [119] created, described and assessed the immunogenicity of gp350-based (gp350D_123_) self-assembled nanoparticles of EBV. They used I3-01 derived from KDPG aldolase and *Aquifex aeolicus’s* LS as self-assembling nanoparticles to display 60 copies of gp350 in a repeating pattern. Nanoparticles carrying the gp350D_123_ generated a greater titer (65 to 133-fold higher) of binding and Nabs in mice and nonhuman primates’ serum than the soluble gp350 monomer. Moreover, they also induced a Th2-biased response in mice. Additionally, gp350D_123_-based nanoparticle vaccinations produced long-lasting antibody responses due to their excellent retention in lymph nodes and thermal stability.

Wichgers Schreur et al. (2021) [121] investigated whether linking antigens to E2p and I3-01 (in parallel to LS), could increase the immunogenicity of a potential subunit vaccine against the zoonotic Rift Valley fever virus (RVFV). To test immunogenicity and effectiveness in vivo, the head domain of RVFV GP Gn, a known target for Nabs, was coupled to these nanoparticles. The Gn head domain, when bound to the LS, decreased mortality in a fatal mouse model and protected lambs from viremia and clinical symptoms following vaccination and experimental challenge. Furthermore, in lambs, the LS nanoparticles, in combination with E2p or I3-01, offered complete protection.

I53-50 with 20 copies of prefusion RSV F GP (a trimeric viral protein (DS-Cav1)) has been investigated as the main target of Nabs (Figure 5I) [129]. These nanoparticles created stable, highly ordered, monodisperse immunogens that show DS-Cav1 at a controlled density and generated 10-fold greater NAbs responses in mice and nonhuman primates than trimeric DS-Cav1. Moreover, Walls et al., 2020 [88], Brouwer et al., 2021 [131] and Arunachalam et al., 2021 [130] exhibited the structure-based design of self-assembling protein nanoparticle immunogens that cause mice to develop strong and protective antibody responses against SARS-CoV-2. In mice, rabbits, and cynomolgus macaques, the nanoparticle vaccinations generated Nab titers that were significantly greater than the prefusion-stabilized spike despite a lower dosage, because they feature 60 copies of monomeric SARS-CoV-2 spike RBDs in a highly immunogenic array (currently in phase 1/2 clinical trials: NCT04742738 and NCT04750343) [88]. They also suggested that these assembled nanoparticles’ excellent yield and stability imply that nanoparticle vaccine production could be scalable to suit mass-production. Interestingly, a recent study evaluated the impact of scaffold-specific antibodies on the antigen-specific immune response using I53-50 [207]. Four different viral antigens, HA (IAV), F (RSV), RBD (SARS-CoV-2) and Env (HIV), were used to monitor the antigen specific response. The scaffold-specific immune response did not affect antigen-specific immune response for HA, F and RBD but reduced Env-specific immune response, since Env is immunologically subdominant to the scaffold. For dominant antigens, prior scaffold immunity does not seem to affect antigen-specific immune responses. Nonetheless, the impact of scaffold immunity will still have to be considered if multiple doses of the same vaccine are administered to the same individuals.

Apart from the self-assembling bacterial proteins described above, supramolecular architectures consisting of different proteins are also attracting attention as candidates for the development of vaccine nanoplatforms. In this regard, a self-assembling foldon, the natural trimerization domain of bacteriophage T4 fibritin and GCN4-based isoleucine zipper from *Saccharomyces cerevisiae,* has been used to facilitate the trimerization of viral structural proteins, such as IAV, HIV-1, RSV, MERS-CoV and SARS-CoV-2 antigens, to mimic their native viral structure [208,209,210,211]. The inclusion of these trimerization domain promotes a stable trimeric structure and has been shown to increase the immunogenicity of antigens, supporting the importance of mimicking native viral trimeric structures [210,212,213,214]. Furthermore, a recent study showed that genetically fusing MERS-CoV RBD with a foldon domain elicited potent RBD-specific neutralizing antibodies and protected hDPP4 transgenic mice from viral infection. Moreover, it has been exhibited that these commonly used protein trimerization domains can be highly immunogenic, but they can be further immunosilenced by the addition of *N*-glycans [215].

## 6. Conclusions

The need to develop safe, scalable, effective, and affordable vaccination strategies is emphasized by the constant appearance and evolution of new pathogens and the lack of vaccines for many infectious diseases. Despite their limited immunogenicity and prerequisite for adjuvants, subunit vaccines offer a safe alternative to vaccine formulations based on live-attenuated and inactivated whole pathogens. Numerous adjuvants have largely non-specific and poorly understood mechanisms of action, and these substances are often associated with undesirable side effects. Therefore, bacterial protein supramolecular assemblies provide an interesting alternative to traditional vaccine adjuvants by combining the characteristics of an immunostimulant with those of an antigen delivery system. Compared to their soluble equivalents, antigenic determinants repeatedly presented on bacterial protein-based nanoparticles show strong immune-activating properties. Important aspects of the immune response, such as uptake by APCs, biodistribution, antigen retention, and activation of TLRs, can be modulated by engineering the self-recognition process at the molecular level. This allows for control of the size, morphology, surface chemistry, and symmetry of the nanoassemblies. Vaccination with bacterial protein-based nanoparticles was shown to promote the production of high levels of antigen-specific antibodies and, often, of a robust cellular immune response, while being safe. Future studies are still required to precisely elucidate how nanoparticles’ shape, supramolecular architecture and surface chemistry affect the interactions with the immune system to fully take advantage of bacterial self-assembling proteins for the delivery of subunit antigens.

## Figures and Tables

**Figure 1 vaccines-10-01920-f001:**
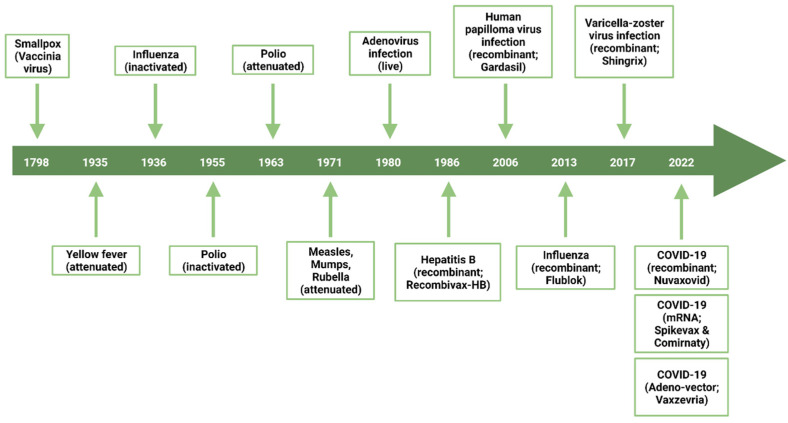
Timeline exposing major breakthroughs related to vaccine development for human uses.

**Figure 2 vaccines-10-01920-f002:**
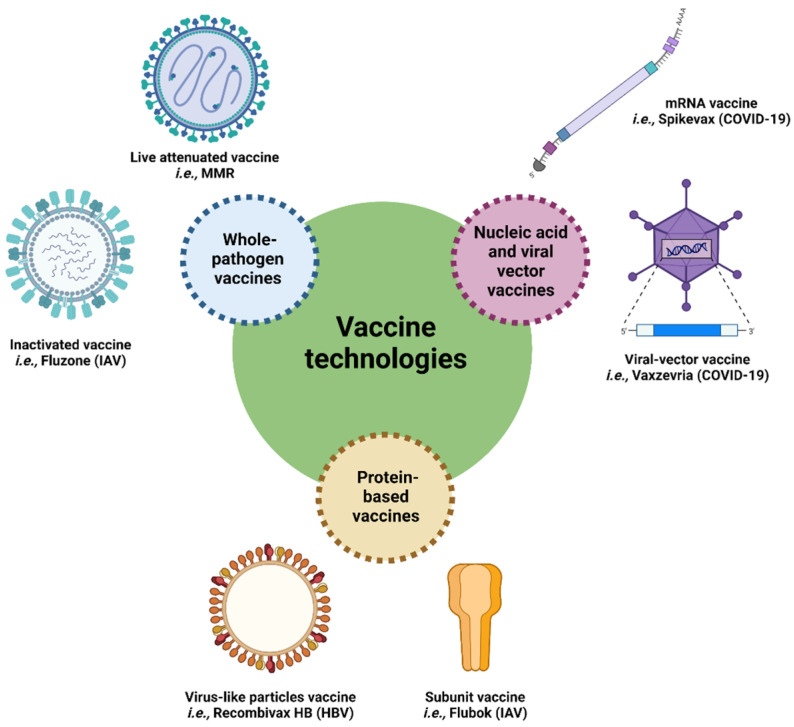
Schematic representation of vaccine technologies and examples of commercialized vaccines for human use. HBV: hepatitis B virus; IAV: influenza A virus; MMR: measles, mumps, and rubella.

**Figure 3 vaccines-10-01920-f003:**
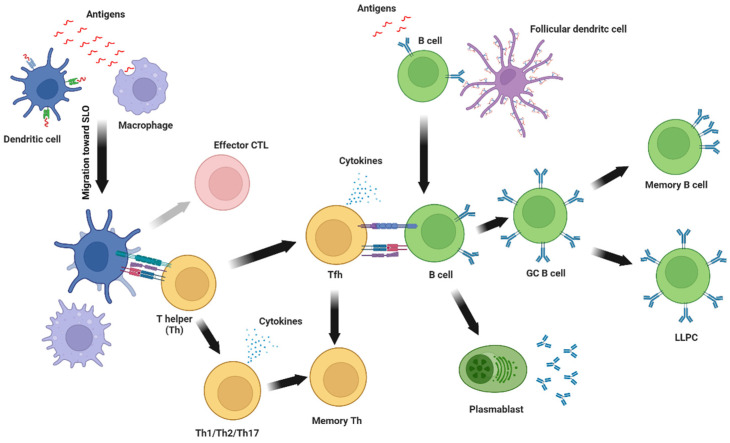
Immune response to subunit vaccines. In brief, antigen presenting cells, including dendritic cells and macrophages, internalize antigens at the site of injection and migrate toward secondary lymphoid organs (SLO) where they present antigen-derived peptides loaded on major histocompatibility complex (MHC) molecules to T helper (Th) and cytotoxic T cells (CTL). Activated Th cellsproliferate and differentiate into Th1, Th2, or Th17 and secrete cytokines and modulate the activity of other immune cells. An activated T helper can also differentiate into T follicular helper (Tfh) which provides direct help for B cell activation and germinal center reaction. Following the contraction of the immune response, some activated T helpers remain as memory T helpers. On the other hand, CTL can be reactivated (grey arrow) following antigen presentation on MHC-I molecules. Simultaneously, antigen-specific B cells are activated by soluble or membrane-bound antigens immobilized on follicular dendritic cells. Activated B cells proliferate and then differentiate into plasmablasts that secrete low-avidity antibodies or become germinal center B cells (GC B cells) where their BCR undergoes somatic hypermutation to increase antibodies’ avidity. GC B cells then differentiate into long-lived plasma cells (LLPC) that constitutively secrete antibodies and reside in the bone marrow or memory B cells that patrol secondary lymphoid organs waiting for subsequent exposure to the same antigen.

**Figure 4 vaccines-10-01920-f004:**
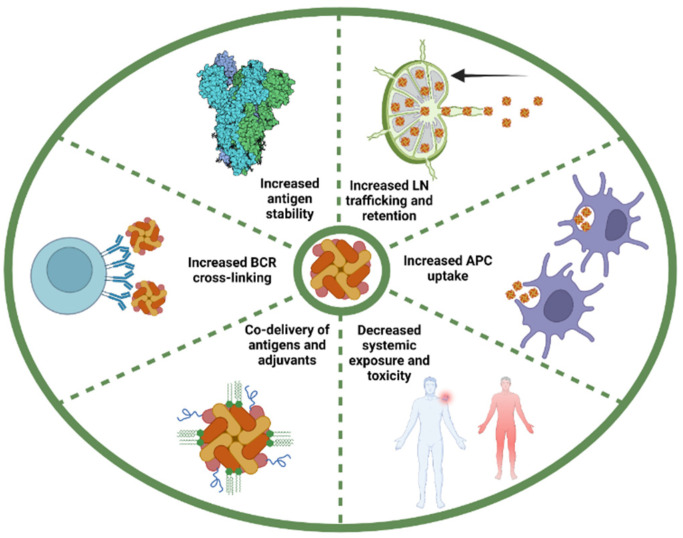
Key advantages of conjugating antigenic determinants on nanoparticle for protein subunit vaccines. Nanoparticle-associated antigens diffuse more efficiently to the draining lymph node (LN) from the injection site and are retained for a longer period of time compared to soluble antigens. The repetitive nature of antigens arrayed on nanoparticles also enhances internalization by APCs via multiple mechanisms. Nanoparticles have also shown to offer limited systemic toxicity. The use of nanoparticles can favor the co-delivery of antigen and adjuvant to the same immune cell, which enhances adjuvant effects and limits off-target effects. Multivalent antigens efficiently induce B cell receptor (BCR) cross-linking, which greatly enhances uptake and presentation by high- and low-affinity B cells. The conjugation of antigen on nanoparticles stabilizes them and allows for the display of antigens under “locked” conformation.

**Figure 5 vaccines-10-01920-f005:**
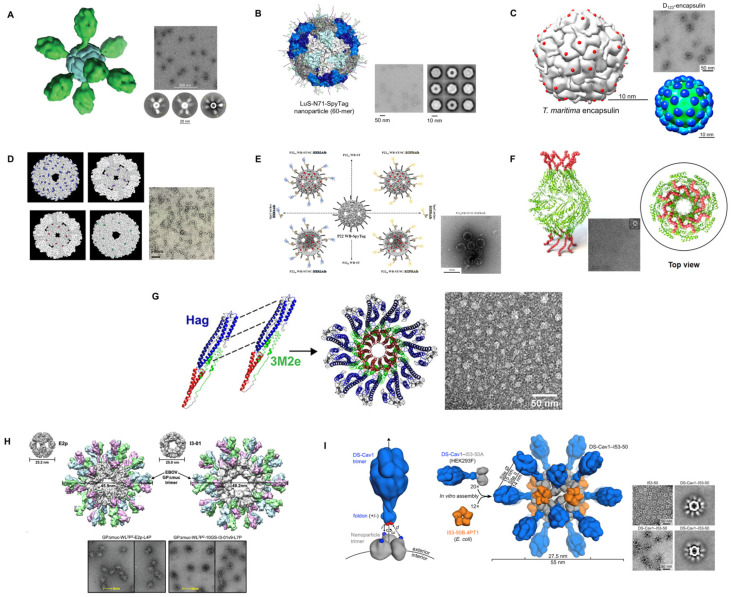
Selected illustrations of self-assembling bacterial proteins as nanoscaffolds for antigen delivery. (**A**) ferritin nanoparticle with the Fusion (RSV) antigen [100]. (**B**) lumazine synthase-N71-SpyTag nanoparticles [89]. (**C**) encapsulin with the insertion sites for exogenous antigens indicated as red spheres [99]. (**D**) Space-filling models of the 24 subunit sHsp cages [132]. (**E**) Schematic representation of various configurations of target-tunable and cargo-loadable P22-based delivery nanoplatforms [133]. (**F**) *Brucella* outer membrane protein BP26-derived nanoarchitecture displaying the influenza extracellular domain of matrix protein-2 (M2e) [126]. (**G**) Hag-3M2e_Ct_ ring-like nanostructures [127]. (**H**) 60-meric E2p and I3-01 with GP (EBOV) trimer [116]. (**I**) Structural model of DS-Cav 1-I53-50 and schematic representation of the self-assembly process. Each nanoparticle comprises 20 trimeric and 12 pentameric building blocks for a total of 60 copies of each subunit [129]. (**A**–**I**) Reprinted with permission from [89,99,100,116,126,127,129,132,133].

**Table 1 vaccines-10-01920-t001:** Expression systems to produce protein subunit vaccines.

Expression System	Advantages	Limitations	Vaccines	Antigens
Bacteria	Simple, well established, low cost, large-scale production	No PTM, inclusions body	Bexsero (against *Neisseria meningitidis*)	fHbp, NadA, NHBA & PorA [24]
Yeast	Simple, low cost, large scale production	Low PTM, hyperglycosylation	All HBV vaccines Gardasil (against HPV) Corbevax (against SARS-CoV-2)	HBs-Ag L1 RBD [24,25]
Insect cells	Human-like PTM, transient Expression	High costs, longer than bacteria and yeast and lower yield	Cervarix (against HPV) Flublok (against IAV) Nuvaxovid (against SARS-CoV-2)	L1 HA S protein [11,26]
Mammalian cells	Human identical PTM, stable expression	High cost, time-consuming to generate stable lines and lower yields	Several candidates against SARS-CoV-2	S, S1 & RBD [21]
Plants	Large scale production, easily modified genome, transient expression	New technology, high time required for implementation	Covifenz (against SARS-CoV-2)	S [27]

PTM: Post-translational modification; fHbp: factor H binding protein; NadA: Neisseria adhesin A; NHBA: Neisserial heparin binding antigen; HBV: Hepatitis B virus; HPV: Human papillomavirus; HBs-Ag: Hepatitis B surface antigen; L1: major capsid protein; IAV: Influenza A virus; S: spike protein; S1: S1 domain of spike protein; RBD: Receptor-binding domain; HA: Haemagglutinin; PorA: Porin A.

**Table 2 vaccines-10-01920-t002:** Bacterial self-assembling proteins evaluated as nanoscaffolds for antigen delivery in subunit vaccines.

Self-Assembling Protein	Organism	Antigen	Expression System	Assembly Structure	Size	Method for Conjugating the Antigen	Animal Model/Administration Route	Studied Immunity	Comments	Reference
Ferritin	*Helicobacter pylori (H. pylori)*	Spike trimers (SARS-CoV-2)	Mammalian Expi293 cells	24 homologous subunits self-assemble in an octahedral (432) symmetry	-	Genetic	Female C57BL/6 mice IM	Cellular	This study revealed that the spike-ferritin nanoparticle vaccine, combined with a potent adjuvant (ALFQ) effectively engages innate immune cells and enhances Spike-specific Th1 and cytotoxic T-cell response Adjuvant: alhydrogel and ALFQ	[94]
Ferritin	*H. pylori*	RBD and/or heptad repeat (HR) (SARS-CoV-2)	*Escherichia coli* BL21 and FreeStyle CHO-S cells		-	SpyTag and SpyCatcher	Balb/c mice, Transgenic hACE2 mice (C57BL/6) and Rhesus macaques SC	Humoral and Cellular	RBD-ferritin or RBD/HR-Ferritin induced stronger NAbs and T-cell response compared to monomers with no apparent antibody-dependant enchancement (ADE) Adjuvant: Sigma adjuvant system (SAS)	[95]
Ferritin	*H. pylori*	Spike trimers (SARS-CoV-2)	Expi293F cells			Genetic	Chinese-origin Rhesus macaques IM	Humoral and Cellular	Ferritin nanoparticles exposing Spike Trimers induced potent humoral and cell-mediated immune responses translated into rapid elimination of replicating virus in the upper and lower airways and lung parenchyma of nonhuman primates following high-dose SARS-CoV-2 respiratory challenge Adjuvant: ACFQ	[96]
Ferritin	*H. pylori*	HA trimers (IAV)	293F cells			Genetic	Balb/C mice and Fitch Ferret IM	Humoral	The ferritin nanoparticles presented 8 trimers of HA and increased the breadth of the humoral immune response to HA stem and RBS Adjuvant: Ribi adjuvant system	[97]
Ferritin	*H. pylori*	H1 HA stem (IAV)	freestyle HEK 293 or HEK 293 MGAT1 cells			Genetic	Balb/C mice and Fitch Ferret IM	Humoral	Vaccination of mice and ferrets with H1–SS-ferritin nanoparticles elicited cross-reactive antibodies that completely protected mice and partially protected ferrets against lethal heterosubtypic H5N1 influenza virus challenge despite the absence of detectable H5N1 neutralizing activity in vitro Adjuvant: SAS	[98]
Ferritin	*H. pylori*-bullfrog hybrid ferritin	GP350 (EBV)	FreeStyle 293F or Expi293F cells		~20–30 nm	Genetic	Mice and Rhesus macaques IM	Humoral	The structurally designed GP350-ferritin nanoparticle vaccine increased neutralization from 10- to 100-fold compared to soluble gp350 by increasing the antibodies directed toward a functionally conserved site of vulnerability, improving vaccine-induced protection Adjuvant: SAS	[99]
Ferritin	*H. pylori*	prefusion F protein trimers (RSV)	293EXPI and CHO cells		20 nm in diameter	Genetic	Balb/C Mice and Rhesus macaques IM	Humoral	The ferritin nanoparticles displayed 8 trimers of perfusion stabilized F protein and increased the generation of NAbs compared to soluble prefusion F trimers. Adjuvant: AF03	[100]
Ferritin	*H. pylori*	E1 and E2 antigenic sequences (HCV)	HEK293F cells			Genetic	in vitro serum binding	N/A	The research group investigated a “multivalent scaffolding” approach by displaying 24 copies of an epitope scaffold on a self-assembling nanoparticle, which markedly increased the avidity of antibody binding	[101]
Ferritin	*H. pylori*	V1V2, gp120 and gp140 trimers (HIV)	N-acetylglucosaminyltransferase I-negative (GnTI/) HEK293S, HEK293F and ExpiCHO cells			Genetic	in vitro antibody binding and B cell activation	N/A	Ferritin nanoparticles displaying trimeric V1V2, gp120 and gp140. Demonstrated high-yield gp140 nanoparticle production and robust stimulation of B cells carrying cognate VRC01 receptors by gp120 and gp140 nanoparticles	[102]
Ferritin	*H. pylori*	Envelope trimers (BG505 SOSIP.664) (HIV)	293F cells		30–40 nm in diameter	Genetic	Balb/c mice and New Zealand White Rabbits IM	Humoral	HIV-1 envelope GP trimers (BG505 SOSIP.664) -bearing nanoparticles were significantly more immunogenic than trimers in both mice and rabbits Adjuvant: MPLA liposomes	[103]
Ferritin	*H. pylori*	Envelope trimers (ConM) (HIV)	293F cells		30–40 nm in diameter	Genetic	New Zealand White Rabbits and Rhesus macaques IM	Humoral	The ConM trimers elicited strong NAb responses against the autologous virus in rabbits and macaques that are significantly enhanced when it is presented on Ferritin nanoparticles Adjuvant: Iscomatrix (Isco) or MF59	[104]
Ferritin	*H. pylori*	Envelope trimers (ConM SOSIP.v7) (HIV)	293F cells			Genetic	New Zealand White Rabbits IM	Humoral	Stronger NAbs responses were elicited when the ConM SOSIP trimers were presented on Ferritin nanoparticles Adjuvant: Squalene emulsion and MPLA liposomes	[105]
Ferritin	*H. pylori*	VP6 (Rotavirus A)	*E. coli* BL21 (DE3) cells and transgenically expressed in the milk of mice		~ 20 nm	Genetic	Balb/c mice PO	Humoral	Recombinant VP6–ferritin nanoparticle vaccine efficiently prevented the death and malnutrition induced by the rotavirus infection in pups Adjuvant: Cholera toxin subunit B (CTB)	[106]
Ferritin	*H. pylori*	GP5 (PRRSV)	Sf9 cells			Genetic	Pigs IM	Humoral and cellular	Immunization with PRRSV modified GP5 protein coupled to ferritin elicited improved protective immunity against PRRSV compared to inactivated vaccine	[107]
Ferritin	*H. pylori*	VP1 & G-H loop (FMDV)	Sf9 cells			Genetic	C57BL/6 mice IM	Humoral and cellular	Ferritin nanoparticles carrying recombinant proteins exhibited good immunogenicity with 66.7% survival rate but less than inactivated vaccine Adjuvant: Montanide ISA201VG	[108]
Ferritin	*H. pylori*	E2 (CFSV)	Sf9 cells			Genetic	Rabbit IM	Humoral and cellular	E2-expressing ferritin nanoparticles induced stronger immune responses than E2 alone Adjuvant: Montanide gel 02	[109]
Ferritin	*H. pylori*	Spike (SARS-CoV-2)	Expi 293 cells		15–19 nm	Spy tag and Spy catcher	Balb/C mice IM	Humoral	Recombinant expression of ferritin with a N-linked glycan increased yield in mammalian expression systems and Increased S-directed Nabs Adjuvant: SAS	[89]
Ferritin	*H. pylori*	RBD & Spike (SARS-CoV-2)	ExpiCHO cells		47.9 nm	Spy tag and Spy catcher	Balb/C mice IP	Cellular and Humoral	The 24-meric RBD-ferritin and spike-ferritin elicited a more potent Nab response than the RBD or Spike alone Adjuvant: MF59 or Alum	[110]
Ferritin	*Escherichia coli (E. coli)*	RBD (MERS-CoV)	*E. coli* strain SHuffle^®^ T7		20–40 nm	Genetic	Balb/c mice IM	Humoral	ChapeRNA-mediated folding of RBD-ferritin controlled the overall kinetic network of the antigen folding pathway in favor of enhanced assemblage of NPs into highly regular and immunologically relevant conformations Adjuvant: MF59 or Alum	[111]
Ferritin	*Pyrococcus furiosus (P. furiosus)*	MD39 env trimer (HIV)	FreeStyle™ 293-F Cells		~40 nm diameter	Genetic	Balb/c mice SC	Humoral	Nanoparticles with heavily glycosylated antigens were accumulated and were retained on FDCs in a mannose-binding lectin- and complement-dependent manner	[112]
Ferritin	*P. furiosus*	preS1 domain of HBV	BL21 (DE3) competent *E. coli*		-	SpyTag and SpyCatcher	Balb/c mice SC	Humoral	preS1-Ferritin nanoparticle targets SIGNR1+ APC, which are involved in Tfh and B cell activation. The vaccine induced a high-level and persistent anti-preS1 response that resulted in efficient viral clearance and partial serological conversion in a chronic HBV mouse model offering a promising translatable vaccination strategy for the functional cure of chronic hepatitis B	[113]
Ferritin	*P. furiosus*	RBD (SARS-CoV-2)	BL21 (DE3) competent *E. coli* and 293F cells			SpyTag and SpyCatcher	C57BL/6 mice SC	Humoral	Vaccine generated an effective antibody response and long-term MBCs in mice that was sustained for at least 7 months after inoculation	[114]
Ferritin	*P. furiosus*	HPV minor capsid protein L2	Sf9 and High Five insect cells	an octahedral structure composed by 24 protomers		Genetic	Balb/c mice and Guinea Pig IM	Humoral	The ferritin-Trx-L2 trimer induced a broadly Nab response covering 14 oncogenic and two non-oncogenic HPV types, which lasted for at least one year Adjuvant: MF59 or Alum	[115]
Ferritin	*Thermotoga maritima (T. maritima)*	GPΔMUC trimer (EBOV)	ExpiCHO	24-subunit protein icosahedron	34.6 nm	Genetic	Balb/C mice IPNew Zealand white rabbit IM	Humoral and Cellular	GP trimers and nanoparticles elicited cross-ebolavirus NAbs, as well as non-NAbs that enhanced pseudovirus infection Adjuvant: MF59 or Alum	[116]
Lumazine synthase	*Aquifex aelocus (A. aelocus)*	gp120 (HIV)	N-acetylglucosaminyltransferase I-negative (GnTI/) HEK293S, HEK293F and ExpiCHO cells	Self-assembles into a 60-mer		Genetic	in vitro BCR expressing cell stimulation	Humoral	Demonstrated high yield gp140 nanoparticle production and robust stimulation of B cells carrying cognate VRC01 receptors by gp120 and gp140 nanoparticles	[102]
Lumazine syn-thase	*A. aelocus*	gp120 (HIV)	FreeStyle™ 293-F Cells		~32 nm diameter	Genetic	Balb/c mice SC	Humoral	The findings highlighted how the innate immune system recognizes HIV nanoparticles and the importance of antigen glycosylation in the design of next-generation nano-based vaccines	[112]
Lumazine syn-thase	*A. aelocus*	gp120 (HIV)	FreeStyle™ 293-F Cells			Genetic	Balb/c mice IP or SC	Humoral	The results suggested that rational epitope design can prime rare B cell precursors for affinity maturation to desired targets Adjuvant: Ribi, Alum or Isco	[117]
Lumazine syn-thase	*A. aelocus*	gp120 (HIV)	FreeStyle™ 293-F Cells			Genetic	Balb/c mice IP or SC	Humoral	When multimerized on nanoparticles, the immunogen (eOD-GT6) activated germline and mature VRC01-class B cells Adjuvant: Ribi, Alum or Isco	[118]
Lumazine syn-thase	*A. aelocus*	Spike (SARS-CoV-2)	Expi 293 cells		15–19 nm	Spy tag and Spy catcher	Balb/C mice IM	Humoral	SARS-CoV-2-spike nanoparticles elicited substantially higher Nab responses than spike alone Adjuvant: SAS	[89]
Lumazine syn-thase	*A. aelocus*	GP350 (EBV)	High-Five cells		~20 nm	Genetic	Balb/C mice SC non-human primate IM	Humoral	Nanoparticle vaccine elicited potent Nab antibody responses against EBV infection Adjuvant: MF59 or Alum	[119]
Lumazine syn-thase	*A. aelocus*	Gc env (SBV)	*E.Coli* BL21 (DE3) and Drosophila S2 Cells		15 nm	Spy tag and Spy catcher	C57BL/6 mice and cattle (German domestic cow breed) SC	Humoral	Even a single-shot vaccination protected about 80% of mice from an otherwise lethal dose of SBV and induced a virtually sterile immunity in cattle Adjuvant: Emulsigen (mice) or Ploygen (Cattle)	[120]
Lumazine syn-thase	*A. aelocus*	Gn (RVFV)	*E. coli* BL21 (DE3) and High FIve cells	Assembles via 12 pentamers into an icosahedral particle	15 nm	Spy tag and Spy catcher	Balb/C mice and Texel-German lamb IM	Humoral	Lumazine synthase-based nanoparticles, prevented mortality in a lethal mouse model and protected lambs Adjuvant: Stimune (mice) and TS6 (lamb)	[121]
Encapsulin	*T. maritima*	M2e (IAV)	*E. coli* BL21 (DE3)	60-mer (T = 1) icosohedral capsid-like particles	24 nm	Genetic	Balb/C mice SC	Humoral	Nanoparticle immunization elicited antibody responses against both the surface epitope and the loaded cargo protein Adjuvant: Freund’s adjuvant	[93]
Encap-sulin	*T. maritima*	GP350 (EBV)	Expi293F		~20–30 nm	Genetic	Balb/C Mice and Rhesus macaques IM	Humoral	The structurally designed nanoparticle vaccine increased neutralization from 10- to 100-fold compared to soluble gp350 by targeting a functionally conserved site of vulnerability, improving vaccine-induced protection in a EBV mouse experimental challenge Adjuvant: SAS	[99]
small Heat shock protein (sHsp) 16.5	*Methanocaldoccus jannaschii (archea)*	Model antigen, ovalbumin (OVA)	*E. coli* BL21 DE3	24 repeating subunits self-assemble to produce cage-like nanoparticles	30–41 nm	Chemically	Balb/C and C57BL/6 mice SC	Humoral	sHsp nanoparticles elicited quick and intense antibody responses, and these accelerated responses could similarly be targeted toward antigens chemically conjugated to the sHsp Adjuvant: Alum	[122]
small Heat shock protein (sHsp) 16.5	*Methanocaldoccus jannaschii*	-	*E. coli*			-	C57BL/6 (CD45.2), BALB/c, and µMT (B10.129S2(B6)-Igh-6^tmlCgn^) mice IN	-	Bronchus-associated lymphoid tissue elicited by a protein cage nanoparticle enhanced protection in mice against diverse respiratory viruses	[123]
P22	Bacteriophage	Model antigen, ovalbumin (OVA)	*E. coli* BL21 DE3	Non-infectious empty viral capsid	30–41 nm	Priming agent	Balb/C and C57BL/6 mice SC	Humoral	Pretreatment of mice with P22 further accelerated the onset of the antibody response to OVA–sHsp, demonstrating the utility of conjugating antigens to VLPs for pre-, or possibly post-exposure prophylaxis of lung, all without the need for adjuvant Adjuvant: Alum	[122]
P22	Bacteriophage	Conserved nucleoprotein (NP) from influenza (H1N1 and H3N2)	*E. coli* BL21 DE3		29–54 nm	Priming agent	BALB/c mice IN	Humoral and Cellular	P22 encapsulating NP (truncated and full-length constructs) elicited a strong protective immune response in mice against challenge with both H1N1 and H3N2 (IAV), without the addition of adjuvants	[124]
P22	Bacteriophage	HA (PR8 IAV)	ClearColi BL21 (λDE3)		26 nm	Spy tag and Spy catcher	C57BL/6 (CD45.2) mice IN	Humoral	P22 VLPs can be rapidly modified in a modular fashion, resulting in an effective vaccine construct that can generate protective immunity without the need for additional adjuvants	[125]
BP26	*Brucella abortus*	M2e (IAV)	*E. coli* BL21 (DE3)	Nanobarrels (forms a barrel-like structure with a hollow center through self-assembly of 16 monomeric proteins)	11–22 nm	Genetic	Balb/C mice SC	Humoral and Cellular	BP26-M2e nanobarrels effectively protected mice from (IAV) infection-associated death, even without the use of a conventional adjuvant Adjuvant: Alum	[126]
Flagellin	*Bacillus subtilis*	M2e (IAV)	*E. coli* Rosetta DE3	Ring-like nanostructures	10–15 nm	Genetic	Balb/C Mice IN	Humoral and Cellular	Flagellin ring-like nanostructures were efficiently internalized by APCs, and avidly activated the TLR5 in vitro as well as the innate and adaptive immune responses	[127]
Flagellin	Salmonella serovar enterica typhimurium (*S. typhimurium*)	Viral envelope protein from Dengue virus (DENV2)	SF9 & SF21 insect cells	Filaments	35 nm	Genetic	C57BL/6J, B10.D2, 6.5-TCR (Tg(Tcra/Tcrb)1Vbo) mice IN or IP	Humoral and Cellular	Reengineered hybrid FliC enhanced T-cell-dependent and possibly induced T-independent antibody responses from B-1 B cells	[128]
E2p	*Bacillus stearothermophilus (B. stearothermophilus)*	gp120/gp140 (HIV)	N-acetylglucosaminyltransferase I-negative (GnTI/) HEK293S, HEK293F and ExpiCHO cells	60-mer assembles into a pentagonal dodecahedral scaffold		Genetic	in vitro BCR expressing cell stimulation	Humoral	Demonstrated high-yield gp140 nanoparticle production and robust stimulation of B cells carrying cognate VRC01 receptors by gp120 and gp140 nanoparticles	[102]
E2p	*B. stearothermophilus*	GP (EBOV)	HEK293F and ExpiCHO cells		45.9 nm	Genetic	Balb/C mice IP New Zealand white rabbit IM	Humoral and Cellular	GP trimers and nanoparticles elicited cross-ebolavirus NAbs, as well as non-NAbs that enhanced pseudovirus infection Adjuvant: MF59 or Alum	[116]
E2p	*B. stearothermophilus*	RBD & Spike (SARS-CoV-2)	ExpiCHO cells		55.9 nm	Spy tag and Spy catcher	Balb/C mice IP	Humoral and Cellular	E2 elicited up to 10-fold higher NAb titers. Adjuvant: MF59 or Alum	[110]
E2p	*B. stearothermophilus*	Gn (RVFV)	E.Coli BL21 (DE3) and High FIve cells		27 nm	Spy tag and Spy catcher	Balb/C mice and Texel-German lamb IM	Humoral	Geobacillus stearothermophilus E2p or a modified KDPG Aldolase provided complete protection in lambs from RVFV challenge Adjuvant: Stimune (mice) and TS6 (lamb)	[121]
I3-01	*T. maritima*	GP (EBOV)	ExpiCHO	60-subunit protein icosahedron	49.2 nm	Genetic	Balb/C mice IP and New Zealand white rabbit IM	Humoral and Cellular	GP trimers and nanoparticles elicited cross-ebolavirus NAbs, as well as non-NAbs that enhanced pseudovirus infection Adjuvant: MF59 or Alum	[116]
I3-01	*T. maritima*	Gn (RVFV)	*E. coli* BL21 (DE3) and High FIve cells		25 nm	Spy tag and Spy catcher	Balb/C mice and Texel-German lamb IM	Humoral	I3-01 modified KDPG Aldolase provided complete protection in lambs from RVFV challenge Adjuvant: Stimune (mice) and TS6 (lamb)	[121]
I3-01	*T. maritima*	GP350 (EBV)	High-Five cells		~25 nm	Genetic	Balb/C mice SC	Humoral	The self-assembled nanoparticle vaccine elicited potent Nabs responses against EBV infection Adjuvant: MF59 or Alum	[119]
I3-01	*T. maritima*	RBD and Spike (SARS-CoV-2)	ExpiCHO cells		59.3 nm	Spy tag and Spy catcher	Balb/C mice IP	Humoral and Cellular	I3-01v9 60-mers elicited up to 10-fold higher NAb titers. I3-01v9 SApNP also induced critically needed T cell immunity Adjuvant: MF59 or Alum	[110]
I3-01	*T. maritima*	GP (EBOV)	HEK293F and ExpiCHO cells		49.2 nm	Genetic	Balb/C mice and New Zealand white rabbit IM	Humoral and Cellular	GP trimers and nanoparticles elicited cross-ebolavirus NAbs, as well as non-NAbs that enhanced pseudovirus infection Adjuvant: MF59 or Alum	[116]
I53-50	*-*	Fusion (RSV)	HEK293F and Expi293 cells	Icosahedral assembly	55 nm	Genetic	Balb/C mice and Indian Rhesus macaques IM	Humoral and Cellular	Computationally designed self-assembling nanoparticle that displayed 20 copies of a trimeric viral protein induced potent Nab responses Adjuvant: MF59 (mice) and Squalene emulsion (macaques)	[129]
I53-50	*-*	RBD of Spike (SARS-Cov-2)	Expi293F		28 nm	Genetic	Balb/C mice and Pigtail Macaque IM	Humoral	The nanoparticle vaccine exhibits 60 copies of the RBD of the spike protein and induced strong humoral response in mice and non-human primates Adjuvant: MF59	[88]
I53-50	*-*	RBD of Spike (SARS-Cov-2)	Expi293F		28 nm	Genetic	Rhesus Macaque IM	Humoral and Cellular	Follow-up study to Walls et al., 2020 [88] where the platform was combined to different adjuvant. All combination showed potent humoral response, but combination with AS03 was the only one that elicited mixed TH1/TH2 cellular response Adjuvant: AS03 or AS37	[130]
I53-50	*-*	Spike (SARS-Cov-2)	HEK293F		30 nm	Genetic	Balb/C mice SC, New Zealand White rabbit IM and Cynomolgus macaque IN	Humoral and Cellular	The I53-50 nanoparticle construct exposed several Spike proteins and induced strong Nabs in all animal models. Furthermore, the vaccine generated IFN-γ secreting T cells in non-human primate Adjuvant: Poly-IC (mice), Squalene emulsion (rabbit) or MPLA liposomes (macaque)	[131]

Administration routes: IM: intramuscular, IN: intranasal, IP: intraperitoneal, PO: oral, SC: subcutaneous.

## Data Availability

No new data reported.

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
