# Peer review of "Vaccination Strategies Based on Bacterial Self-Assembling Proteins as Antigen Delivery Nanoscaffolds"

_vaccines, 2022, doi:10.3390/vaccines10111920_

Round 1
Reviewer 1 Report
In this review, the authors provide a nice and extensive summary of how bacterial nanoparticle-based scaffolds were applied as antigen-delivery vehicles for vaccines. Several commonly used nanoparticle scaffolds were used as examples to demonstrate the advantages of this platform. However, there are still some important questions that remain unanswered.
1. Please provide high-resolution figures.
2. sections 2-3 and 4.1/4.2 are a bit wordy, most of the contents are basic immunology and are not closely related to the topic of this review paper – self-assembling antigen delivery scaffolds. I would recommend making these sections more concise, and focusing more on the following questions in comments 4 & 5.
3. Table 2: Please add a column for the adjuvants used in these studies, if there are any.
4. The immunization regimen is also an important factor that dictates the outcome of the vaccines. I wonder compared to regular vaccines, do nanoparticle-based vaccines generally require less dosage (amount of protein) or fewer boosters to induce similar levels of immune responses? What injection routes are usually used to administer nanoparticle-based vaccines?
5. Will there be off-target effects that the immune responses are biased to the scaffold instead of the target antigen? Are there other potential limitations or caveats of nanoparticle-based vaccines? It would be nice to include some more thoughts to answer these questions.
Reviewer 2 Report
The aim of this review is to contribute in the knowledge of bacterial self-assembling proteins as promise tool for the improvement of the immune protection against infection diseases. In this sense, firstly the authors show an overview of the relevance of the vaccination for the public health, as well as a timeline of the approved vaccines and the main strategies of the commercial vaccines. Furthermore, the review also reinforces the mechanisms of the immune activation and induction of the immune protection against pathogenic organisms. Afterwards, the authors also describe several scientific reports of structural characteristics of bacterial self-assembling proteins and potential role for induction pathogen-specific protective immune responses in distinct models. Therefore, the review presents an overview of vaccinology with focus on bacterial self-assembling proteins. Despite that, some points should be better discussed or altered to improve the impact of the review.
The authors should check the references that are cited, as examples:
Line 46 - The reference cited by the authors is not the better to support the sentence. The focus of the cited reference: “A biomimetic VLP influenza vaccine with interior NP/exterior M2e antigens constructed through a temperature shift-based encapsulation strategy”.
Line 49- The authors were generic for the efficiency of the influenza vaccine around 19% in the sentence, the data showed in the cited report should be detailed.
The authors should include some reports about:
Inflammasome and vaccines,
Efficiency of vaccines in the context of immunosuppression and imunodeficiencies.
The figure 4 should be explored in details
sub-item: costs/infrastructure and advantages for the production of different vaccine strategies.
The references must be revised.
Reviewer 3 Report
Overall, this manuscript presents a review of vaccine types, a summary of immune responses to vaccines, description of adjuvants, and ultimately a future focus on bacterial particulate antigen presentation. While the sections are carefully researched and quite detailed, one gets the impression that the authors lack direct hands-on experience with animal or human models of immune responses to viral infections and vaccines (and associated literature) and thus make some critical conceptual errors (see below). As an immunologist who has measured CD4 and CD8 T-cell and antibody immune responses to virus preparations (peptides, proteins, killed and live virions) in mice and humans, I have a great deal of appreciation as to how particulate antigen forms induce a much stronger and diverse functional response from all cell types and found the discussion of bacterial self assembling proteins and nanoparticles informative and of great interest. Also, as an immunologist who has hands on experience harvesting T and B cells from spleens and lymph nodes of mice immunized with subunit vaccines and adjuvants (allum and CFA), in which the adjuvants cause extensive adhesions and fibrosis, I am greatly in favor of developing particulate vaccine forms that do not require adjuvants. The discussion as to the importance of PTM’s depending on the expression system for proteins is also of considerable interest, not only in terms of vaccine design, but also as it relates to measurement of antibody responses to infections and vaccinations.
1) This reviewer’s major issue with this manuscript (shown in figure 3) is that the authors propose that a subunit vaccine can elicit a primary cytotoxic T-cell response to virus infected cells. Unfortunately, all our knowledge suggests this is not the case. Subunit and inactivated viral vaccine preparations do not generate a cytotoxic T-cell response because a primary CD8 CTL response requires that the viral protein be synthesized in the host cells for appropriate presentation with HLA-class I. This well accepted finding was/is a major driver of nucleic acid based vaccines as well as the continued use of attenuated viruses and viral vectors. While it is possible for dendritic cells (DC) to “cross present” to CD8 CTL as shown in figure 3, cross presentation appears to be rare In Vivo and requires that autologous DC be loaded with HLA matched viral epitopes In Vitro and there is a whole literature on DC based vaccines. Because of this limitation, it is likely that the self folding bacterial particles proposed will also not generate CD8 CTL. This is not necessarily the kiss of death for the technology since CTL do not always play a critical role in defense against extracellular bacterial infections and are less important with viral infections such as hepatitis in which the viral burden is primarily extracellular and sensitive to antibodies. Also, while they may not be able to establish a primary CTL response, protein subunit vaccines in particulate form do expand memory CD8 T –cells and are thus useful with commonly encountered pathogens such as influenza.
2) Monocytes and macrophages are more numerous than DC and likely play a large role in antigen presentation In Vivo and yet are barely discussed in this the report.
Monocyte/macrophages rapidly phagocytose particulate antigens such as virions, bacterial and protein complexes and likely the self assembling bacterial proteins discussed. In response to particulate (but rarely soluble peptide or protein antigens)myloid cells synthesize a broad array of inflammatory proteins. In contrast, DC are rare and there are many different types and maturational stages. The more mature forms of DC are poor at antigen antigen processing, so if the authors are interested in particulate antigens such as the self assembling bacterial proteins, they might want to discuss monocyte/macrophage uptake of particles and responses.
Reviewer 4 Report
great review, well written.
-Question: Should fibritin and foldon carrier from bacteriophage be briefly discussed?
-Please replace COVID-19 by SARS-CoV-2 where appropriate as you are actually referring to the virus, not the disease it causes (text, all tables and figures). Rephrase 142-144. Same comment for meningitis in table 1.
-line 163 invert B and T.
-add macrophages to figure 3 with resident dentritic cells and FDC
-line 240: respectively?
-line 341: cellular and humoral
-Suggestion for table 2: group vaccines based on the antigen and not the organism? The comments section could also be shortened for some vaccines.
-remove viral line 521
Round 2
Reviewer 1 Report
All the previous comments were well addressed.
Reviewer 2 Report
The revised manuscript has been changed according to the main suggestions.
Reviewer 3 Report
The authors have addressed all concerns and improved the manuscript.